# Validation of an HPLC Method for the Simultaneous Quantification of Metabolic Reaction Products Catalysed by CYP2C11 Enzymes in Rat Liver Microsomes: In Vitro Inhibitory Effect of Salicylic Acid on CYP2C11 Enzyme

**DOI:** 10.3390/molecules24234294

**Published:** 2019-11-25

**Authors:** Hassan Salhab, Declan P. Naughton, James Barker

**Affiliations:** School of Life Sciences, Pharmacy and Chemistry, Kingston University, Kingston upon Thames, London KT1 2EE, UK; d.naughton@kingston.ac.uk (D.P.N.); j.barker@kingston.ac.uk (J.B.)

**Keywords:** cytochrome P450, HPLC, 16α-hydroxytestosterone, phenacetin, salicylic acid, testosterone

## Abstract

The inhibitory effect of new chemical entities on rat liver P450 marker activities was investigated in a functional approach towards drug development. Treatment of colorectal cancer (CRC) and chemoprevention using salicylic acid has gained a lot of attention, mainly in the prevention of the onset of colon cancer. Thus, an in vitro inhibitory effect of salicylic acid on rat CYP2C11 activity was examined by using high performance liquid chromatography (HPLC). High performance liquid chromatography analysis of a CYP2C11 assay was developed on a reversed phase C_18_ column (SUPELCO 25 cm × 4.6 mm × 5 µm) at 243 nm using 32% phosphate buffer (pH 3.36) and 68% methanol as a mobile phase. The CYP2C11 assay showed good linearity for all components (R^2^ > 0.999). Substrates and metabolites were found to be stable for up to 72 h. Additionally, the method demonstrated good reproducibility, intra- and inter-day precision (<15%), acceptable recovery and accuracy (80%–120%), and low detection (1.3501 µM and 3.2757 µM) and quantitation limit values (4.914 µM and 9.927 µM) for 16α-hydroxytestosterone and testosterone, respectively. Salicylic acid acts reversibly as a noncompetitive (weak) inhibitor with K_i_ = 84.582 ± 2.67 µM (concentration of inhibitor to cause 50% inhibition of original enzyme activity (IC_50_) = 82.70 ± 2.67 µM) for CYP2C11 enzyme activity. This indicates a low potential to cause toxicity and drug–drug interactions.

## 1. Introduction

Cytochrome P450s are known as Phase 1 mono-oxygenase drug-mediated enzymes that play a vital role as a body defence in the drug pharmacokinetic field [1]. Cytochrome P450s have the ability to transform biologically xenobiotic drugs, such as steroids, fatty acid derivatives, hormones, and endogenous and exogenous molecules, into more lipophobic molecules (more polar functional groups) [2]. CYP450 drug metabolising enzymes are located predominantly in the liver, whereas a few of them are expressed in other human body organs, such as the small intestine, kidneys, lungs and placenta [3]. Statistical results have shown that about 70%–80% of most marketed drugs are metabolised by five major Phase 1 drug mediated enzymes [4].

Nowadays, toxicity and bioavailability issues are the most common obstacles in the drug development field [4]. Therefore, induction or inhibition of CYP450 enzymes can result in either drug–drug or drug–herb interactions, and their effectiveness can be altered [5]. Thus, this leads to a therapeutic failure in the treatment [3]. Understanding the concept of inhibition and induction pathways can help in reducing the interaction between drugs and occurrence of adverse reactions [3]. Inhibition of CYP450 enzymes by drugs can lead to a rise in the concentration of other metabolising drugs, thus causing drug toxicity problems [6].

The CYP2C11 isoform is the most dominant isoform in male-specific rat liver microsomes and accounts for 50% of the total male-specific male rat hepatocytes [7]. Wojcikowski et al. [7] stated that testosterone is metabolised to either 16α-hydroxytestosterone or 2α-hydroxytestosterone by means of CYP2C11 enzyme activity in male-specific rat liver microsomes. Previous studies showed that some inflammatory mediators, such as ethanol, cimetidine and diclofenac, extensively inhibit the CYP2C11 isoform [7]. Thus, both 2α and 16α-hydroxytestosterone are used as markers in studying CYP2C11 enzyme activity in rat liver microsomes [7].

Salicylic acid (Figure 1) belongs to a nonsteroidal, anti-inflammatory drug class, which plays a vital role in the reduction of gastrointestinal tumours [8]. However, recent studies revealed that salicylic acid can cause liver damage [9]. These studies have shown that salicylic acid can be metabolised to six metabolites using various quantitative analytical techniques, such as HPLC and liquid chromatography tandem mass spectrometry (LC-MS/MS) [10]. Salicylic acid inhibits the metabolism of fatty acid based on the acyl chain length [11]. These studies suggested that salicylic acid can be linked to the interference of fatty acid metabolism by reducing [C14] CO_2_ from fatty acid and increasing the amount of uncoupled fatty acid in the plasma. Until now, various papers on the pharmacokinetics and metabolism of salicylic acid have been reported, but only one paper [9] has shown that the oxidative metabolism of salicylic acid is mediated by CYP2E1 and CYP2C11 enzymes in rat liver microsomes.

Consequently, an analytical HPLC method was validated, which studies the in vitro inhibitory effect of salicylic acid on the activities of major cytochrome CYP450s (CYP2C11) as an example for promoting the safety and efficacy of salicylic acid in clinics. 

## 2. Results and Discussion

### 2.1. Selection of Analytical Wavelength: UV-VIS (Ultraviolet-Visible) Spectroscopy (CYP2C11 Assay)

UV-VIS spectrophotometry analysis for CYP2C11 assay was performed by dissolving salicylic acid (100 µM), testosterone (200 µM) and 16-α hydroxytestosterone (50 µM) powder in pure methanol (wavelength cut-off of methanol was 210 nm).

The following graph illustrates the measurements of maximum wavelength of each component in the CYP2C11 assay. 

From overlain spectra of CYP2C11 assay compounds (Figure 2) and (Appendix A), it is shown that the maximum of the absorption band for all four CYP2C11 components is 243 nm, and so this was chosen for our study.

### 2.2. Method Validation

#### 2.2.1. Linearity and Range

Different concentrations of testosterone and its metabolite (16α-hydroxytestosterone) solutions (mentioned in Section 3.6.1) were analysed by HPLC for the production of standard calibration curves. Calibration curves were constructed by plotting the mean area peak of standards and phenacetin (internal standard) versus the concentration of standard (substrate and metabolite). The outcomes are listed in Table 1. Good linearities for both testosterone and 16α-hydroxytestosterone were obtained, with *r*^2^ values of 0.9999 and 0.9998, respectively. The linear regression coefficient was within the acceptable range (*r*^2^ > 0.99), according to ICH (International Conference on Harmonization) guidelines. Relative standard deviations (%RSD) at each different concentration (<5%) met the ICH guidelines [12].

#### 2.2.2. Limit of Detection (LOD) and Limit of Quantitation (LOQ)

According to ICH Technical Requirements for Registration of Pharmaceuticals for Human Use, the LOD and LOQ of the proposed method were calculated mathematically by the relationship between the slope of the calibration curve and the standard deviation of the response using the following equations
LOD = 3.3σ/S                 LOQ = 10σ/S
where σ is the standard deviation of the response; S is the slope of the calibration curve.

The results presented in Table 2 demonstrate that both testosterone and 16α-hydroxytestosterone have detection values in the range of 1–4 µM and quantitation values in the range of 4–10 µM, which is consistent with the literature [13].

#### 2.2.3. Specificity and Selectivity

Specificity tests were performed by using spiked samples (50 µM of phenacetin dissolved in 30% of phosphate buffer (pH 3.36) + 70% of methanol) to evaluate that the method outcomes were not affected by the impurities. 

Specificity was achieved by choosing the right mobile phase composition (phosphate buffer at pH 3.36 (A) + methanol (B)) using low pressure isocratic elution programming: 68% methanol + 32% phosphate buffer at pH = 3.36. This results in good separation of the enzyme peak (NADPH (Nicotinamide Adenine Dinucleotide Phosphate Hydrogen)-regenerating system) from the CYP2C11 metabolite (16α-hydroxytestosterone), salicylic acid, phenacetin and testosterone peaks at T = 25 °C, using a C18 (SUPELCO 25 cm × 4.6 mm, 5 µm) column at 0.8 mL/min flow rate and a wavelength of λ = 243 nm (Figure 3).

#### 2.2.4. Precision

##### Intra-Assay Variation of Testosterone and 16α-hydroxytestosterone

Intra-assay variation of testosterone and 16α-hydroxytestosterone was determined by measuring three testosterone concentration levels (200, 100, and 25 µM, representing high, medium, and low levels, respectively) and three 16α-hydroxytestosterone concentration levels (80, 40, 10 µM, representing high, medium, and low levels, respectively) 3 times in a single batch (n = 3). The outcomes, summarised in Table 3 and Table 4, illustrate that the %RSD (relative standard deviation) was < 5% for testosterone and 16α-hydroxytestosterone. The experiment revealed that there is no significant variation in intra-assay measurements.

##### Inter-Assay Variation of Testosterone and 16α-hydroxytestosterone

Inter-assay variation was determined by measuring testosterone standards of three concentrations levels (200, 100, and 25 µM, representing high, medium, low levels, respectively) and 16α-hydroxytestosterone (80, 40, 10 µM) for three consecutive days, in separate batches. The results, summarised in Table 5 and Table 6, illustrate that the relative standard deviation or %RSD was <10% for testosterone and 16α-hydroxytestosterone. The experiment revealed that there is no significant variation between aliquots of the same batch sample in inter-assay measurements.

#### 2.2.5. Stability Test

##### Stability Test of Testosterone 

The stability of testosterone was investigated at three different concentrations (25, 100, and 200 µM, representing low, medium and high levels, respectively), which were chosen as an approximation of its K_m_ [14], and stored for 72 h at room temperature in natural light conditions. Phenacetin (as an internal standard of 50 µM) was added to each batch. Samples were analysed in triplicate (n = 3) for each batch. The stability test results are summarised in Table 7.

The outcomes revealed that there was no variation in the concentration or changes in chromatographic behavior of testosterone at 0, 24, 48, and 72 h, compared to actual concentrations. Calibration curves were plotted for days 1−4 and all four calibration curves were averaged, hence the average calibration curve equation was: y = 0.0236x − 0.0406 (R^2^ = 0.9997), where r^2^ met ICH guidelines. LOD and LOQ values were 5.4408 and 16.4874 µM, respectively. Percentage recovery for testosterone at concentrations of 25, 100 and 200 µM was found to be within an acceptable range (80%–120%), according to ICH guidelines, and thus the results show high and acceptable accuracy (80%–120%) for testosterone concentrations of 25 and 100 µM. On the other hand, at 200 µM concentration, the accuracy was just beyond the acceptable value of 120.5% for 0 and 24 h. The results indicate that testosterone solution was relatively stable for 72 h at ambient temperature, which is consistent with the literature [15].

##### Stability Test of 16α-hydroxytestosterone 

Stability of CYP2C11 metabolite (16-alfa hydroxytestosterone) was investigated for three different concentrations (10, 40, and 80 µM) stored for 72 h and kept at room temperature in natural light conditions. Three different concentration ranges of 16-alfa hydroxytestosterone were chosen, since the amount of metabolite formed at 200 µM of testosterone was 35 µM, meaning it was necessary to investigate the stability of the metabolite at low, medium, and high concentrations. Phenacetin (as an internal standard of 50 µM) was added to each batch. The sample was analysed in triplicate (n = 3) for each batch. The stability test results are summarised in Table 8.

The results presented in Table 8 show that there was no variation in the concentration or in changes in chromatographic behaviour of 16-alfa hydroxytestosterone at 0, 24, 48 and 72 h compared to actual concentrations. Calibration curves were plotted for days 1–4 and all four calibration curves were averaged, hence the average calibration curve equation was: y = 0.0182x – 0.0117 (r^2^ = 0.9992), where r^2^ met the ICH guidelines. Percentage recovery for 16-alfa hydroxytestosterone at concentrations of 10, 40 and 80 µM was found to be within acceptable ranges (80%–120%), according to ICH guidelines. Thus, the results show high and acceptable accuracy (80%–120%) for 16-alfa hydroxytestosterone concentrations of 10, 40 and 80 µM compared with the standard known concentration. Therefore, the results indicate that 16-alfa hydroxytestosterone solution was stable for 72 h at room temperature, which is consistent with the literature [16].

#### 2.2.6. Robustness Test

##### Changing the Percentage of the Mobile Phase

A robustness study was performed by changing the percentage of the mobile phase (increasing the percentage of methanol by 2%) using HPLC, and evaluating the effect on retention time and peak area of each compound. Table 9 shows the change in mobile phase ratio on both retention time and peak area of each compound. 

Overall, retention time, peak area and resolution for salicylic acid, phenacetin, testosterone and 16α-hydroxytestosterone did not vary when changing the composition of methanol by 2%, except that testosterone eluted later (t_R_ = 12.359 min) when the concentration of methanol decreased by 2%. This means that testosterone is more soluble in organic solvent than aqueous medium. A mobile phase consisting of 68% methanol with 32% phosphate buffer at pH 3.36 was chosen because the resolution between phenacetin and 16-alfa hydroxytestosterone peaks (1.083 min) was greater than the resolution between phenacetin and 16-alfa hydroxytestosterone peaks (0.842 min) when using 70% methanol with 30% phosphate buffer at pH 3.36. This implies that the CYP2C11 assay method is robust, considering the change in mobile phase ratio.

##### Changing the Column Temperature

A robustness study was conducted by considering the change in column temperature on HPLC, by evaluating the effect on retention time and peak area of each compound. As 30 °C was the optimum column temperature for HPLC, thus 25 °C was selected, as it is ± 5 °C. Table 10 shows the effect of temperature on both retention time and peak area of each compound in CYP2C11 assay.

The peak area and the retention time for each component in the CYP2C11 assay remained unaffected when the temperature was changed from 30 °C to 25 °C. However, testosterone eluted earlier when the temperature increased from 25 °C to 30 °C. A temperature of 25 °C was chosen in this analytical method, because the resolution between salicylic acid and the phenacetin peak (1.287 min) at 25 °C was greater than the resolution between salicylic acid and the phenacetin peak (1.26 min) at 30 °C. Overall, this method is robust, considering the change in column temperature.

### 2.3. Effects of Salicylic acid on CYP2C11 Activity

Different concentrations of testosterone (25, 50, 100, 150 and 200 µM) were incubated in the presence of 0, 50, 100 and 200 µM salicylic acid using 0.5 mg/mL of rat liver microsome with 1.0 mM NADPH (Nicotinamide Adenine Dinucleotide Phosphate Hydrogen), 5.0 mM G6P (Glucose-6-Phosphate), 1.7 units/mL G6PDH (Glucose-6-Phosphate Dehydrogenase), 1.0 mM EDTA (Ethylenediamine tetraacetic acid) and 3.0 mM magnesium chloride and the reaction was terminated at different time intervals. The effect of salicylic acid on CYP2C11 activity is shown in Table 11 and Figure 4 and Figure 5.

The experimental method for a CYP2C11 assay has been validated in this study for testosterone and 16α-hydroxytestosterone. All the analytical parameters (accuracy, precision, % error, % recovery, LOD, LOQ, linear regression) were in line with ICH guidelines.

Our in vitro results demonstrate clearly that salicylic acid, at therapeutically relevant concentrations (0–200 µM) [17], acts potently as a reversibly noncompetitive inhibitor, which may inhibit cytochrome P450 2C11 enzyme activity, since the calculated *K*_m_ (substrate concentration at which the reaction rate is half of its maximal value) remains unaffected at three salicylic acid concentrations (50, 100 and 200 µM). However, the *V*_max_ (maximal rate of the reaction at which enough substrate molecules completely fill the enzyme active sites) for the inhibition studies (50, 100 and 200 µM salicylic acid) decreased compared to the *V*_max_ of the negative control assay, as shown in Table 11 and in the Appendix A (Appendix A, Appendix A and Appendix A). This means that salicylic acid as an inhibitor reduces the activity of the CYP2C11 enzyme and is involved in the binding to an allosteric site. Thus, salicylic acid weakly inhibits the CYP2C11 enzyme activity with K_i_ = 84.58 ± 2.67 µM (concentration of inhibitor to cause 50% inhibition of original enzyme activity (IC_50_) = 82.70 ± 2.67 µM). A higher value of K_i_ for salicylic acid in rat liver microsomes has a potentially negligible effect in causing drug interactions with other co-administrated drugs, which are substrates of the CYP2C11 enzyme. Outcomes from this in vitro study looking at the inhibitory effect of salicylic acid on CYP2C11 enzyme activity will be beneficial for a future in vivo study for healthcare screening for the effective use of salicylic acid in clinics and for the safe administration of salicylic acid with other drugs. As salicylic acid metabolism is mediated by two CYP450 enzymes (CYP2C11 and CYP2E1) in rat liver microsomes, so it will be necessary to evaluate the in vitro and in vivo inhibitory effects of salicylic acid on CYP2E1 enzyme activity in humans in order to assess the full effects.

## 3. Materials and Methods

### 3.1. Materials 

HPLC analytical reagent grade methanol and acetonitrile were purchased from Sigma Aldrich, Co. (Old Brickyard, Gillingham, UK). Salicylic acid, potassium phosphate monobasic, potassium phosphate dibasic, phenacetin with purity greater than 98%, phosphoric acid (85% w/w), glucose-6-phosphate (G-6-P), glucose-6-phosphate dehydrogenase (G-6-PDH), EDTA (Ethylenediamine tetraacetic acid), NADP^+^ (Nicotinamide Adenine Dinucleotide Phosphate), magnesium chloride (MgCl_2_), testosterone, microsomes from liver pooled from male rats (Sprague-Dawley) (Gillingham, UK) and 16-alfa hydroxytestosterone were purchased from Sigma Aldrich, Co (Old Brickyard, Gillingham, UK). 

### 3.2. Instrument

A 570 pH Meter purchased from JENWAY Limited (Beacon Road, Stone, Staffordshire, ST15 0SA, UK) was used. A UV-VIS spectrometry instrument was purchased from VWR International Ltd. (Magna Park, Lutterworth, Leicestershire, LE17 4XN, UK) and UV spectra were obtained using a Bio 100 Cary software from Aglient Technologies LDA (Cheadle Royal Business Park, Stockport, Cheshire, SK8 3GR, UK) and 1 cm length of quartz cuvette. A Shimadzu LC-2010A HT Module liquid chromatographic system was used (Shimadzu, Tokyo, Japan), combined with a low pressure pump quaternary gradient (series 200 LC (Liquid Chromatography) pump), a degasser, a model series 200 UV-detector, a series 200 Peltier LC column oven for chromatographing the analysed solutions, and a series 200 autosampler. A SUPELCO (Fancy Road, Poole, Dorest, BH12 4QH, UK) C18 column (25 cm × 4.6 mm, 5 µm particle size) was used. The data were processed using Shimadzu HPLC 2 data lab solutions software processing system.

### 3.3. CYP450 Assay

#### 3.3.1. CYP2C11 Substrate and Its Metabolite

Validation studies were carried out using a LC-2010A HT Module HPLC system (Shimadzu, Toyko, Japan). Chromatographic experiments were assessed in a low pressure gradient mode. The separation of the four target components (salicylic acid used as a tested inhibitor, phenacetin used as an internal standard, testosterone as CYP2C11 substrate and 16α-hydroxytestosterone as CYP2C11 metabolite) was conducted on a SUPELCO C18 column (25 cm × 4.6 mm, 5 µm particle size). The mobile phase for chromatographic separation of the four compounds consisted of 32% of phosphate buffer solution (pH = 3.36) and 68% of methanol. The flow rate was set to 0.8 mL/min, and the oven temperature was 30 °C. A wavelength detection of 243 nm was used and 10 µL solution volume was injected for HPLC analysis. The mobile phase consisted of methanol/phosphate buffer (pH 3.36) (68%/32% v/v), which provided good separation and resolution for the investigated compounds.

#### 3.3.2. Microsomal Incubations and Treatment Protocol

CYP2C11 activity was determined using testosterone (substrate) via the formation of 16α-hydroxytestosterone (metabolite for CYP2C11 enzyme). Oxidative metabolism of testosterone was measured using a NADPH-regenerating system consisting of: 1.0 mM NADPH (Nicotinamide Adenine Dinucleotide Phosphate Hydrogen), 5 mM G6P (Glucose-6-Phosphate), 1.7 units/mL G6PDH (Glucose-6-Phospahe Dehydrogenase), 1.0 mM EDTA (Ethylenediamine tetraacetic acid), and 3.0 mM magnesium chloride. Mixtures were incubated in a final volume of 500 µL of NADPH-regenerating system and and a final concentration of 0.067 M potassium phosphate buffer at pH 7.4 using a serial range of testosterone (25, 50, 100, 150, and 200 µM, dissolved in mobile phase) and a serial range of salicylic acid solutions (0, 50, 100 and 200 µM, dissolved in mobile phase), which were added to the incubation mixture in triplicate. The reaction was initiated by adding 0.5 mg/mL of pooled liver microsomes to each tube. NADP^+^ (Nicotinamide Adenine Dinucleotide Phosphate) was added to the mixture after pre-incubation of all components for 10 min in a water bath at T = 37 °C [18]. The final concentration of organic solvent did not exceed the 1% volume. Tubes were incubated for 60 min in an Eppendorf thermomixer (Eppendorf UK Limited, Stevenage) at 800 × *g* (37 °C). The reaction was terminated after 65 min by the addition of ice-cold grade acetonitrile containing 50 µM of phenacetin (as an internal standard). Tubes were centrifuged in a microcentrifuge (13,000 × *g*) for 12 min to precipitate protein. Then, the supernatant was collected and dissolved in a mobile phase (30% phosphate buffer at pH 3.36 and 70% methanol) and made up to 1000 µL volume. A volume of 10 µL of dissolved supernatant was injected into the instrument for HPLC analysis. 

### 3.4. Selection of Analytical Wavelength

#### CYP2C11 Assay

Phenacetin (50 µM), salicylic acid (100 µM), testosterone (200 µM), and 16α-hydroxytestosterone (50 µM) standard solutions were recorded in the UV region of 200–350 nm using methanol as a blank, and 243 nm absorption wavelength.

### 3.5. Preparation of Mobile Phase

#### CYP2C11 Assay

Different mobile phases for the CYP2C11 assay were used. Thus, the most suitable mobile phase was as follows: HPLC grade methanol (low UV cut-off of 205 nm) as mobile phase (A), and phosphate buffer at pH = 3.36 as mobile phase (B) (A: 68%, B: 32%). 

### 3.6. Preparation of Standard and Sample Solutions

#### 3.6.1. CYP2C11 Assay

##### Analytes Standard Solution Preparation

Salicylic acid (SA) (1.38 mg) (C = 200 µM) was weighed accurately and dissolved in a 50 mL volumetric flask in a mobile phase (70% methanol + 30% phosphate buffer at pH = 3.36). Serial dilutions were performed, yielding final concentrations of 150, 100, 75, 50, 25, and 10 µM. Testosterone (5.76 mg) (C = 400 µM) was weighed accurately and added to a 50 mL volumetric flask before being dissolved in mobile phase. A serial dilution of testosterone stock solution was made, yielding final concentrations of 300, 200, 150, 100, 50, and 25 µM. Phenacetin was used as an internal standard for the CYP2C11 enzyme assay by dissolving 0.0009 g of the powder in a mobile phase (70% methanol + 30% phosphate buffer at pH = 3.36) and a 100 mL volumetric flask.

##### Metabolite Standard Solution Preparation

The metabolite for the CYP2C11 enzyme (16α-hydroxytestosterone) stock solution of 100 µM (in a 50 mL volumetric flask) was prepared, followed by serial dilutions to 80, 60, 40, 20 and 10 µM respectively.

### 3.7. Data Analysis

The regression equation (standard and calibration curves) consisted of different ranges of testosterone and 16α-hydroxytestosterone concentrations using 50 µM of phenacetin as an internal standard, which was calculated by a weighted least-squares linear regression analysis of mean peak area ratio (peak area of standard/peak area of internal standard) versus standard concentrations. Validation parameters were calculated using Microsoft Excel 2010 software (Microsoft Corp. London, UK).

The CYP inhibition analysis was assessed by measuring the formation of 16α-hydroxytestosterone metabolite of the tested CYP2C11 substrate (testosterone). The peak area ratios of both the metabolite and internal standard were acquired using Microsoft Excel 2010 software. Pharmacokinetic parameter (*V*_m_, *K*_m_, *Cl*_int_, α^,^, *K*_i_) values were obtained from secondary Lineweaver–Burk and Michaelis–Menten plots. Inhibition data of CYP2C11 assays were assumed as non-competitive inhibition based on the shape of Lineweaver–Burk plots, and the standard error. AIC (Akaike information criterion) and SC (Schwarz criterion) were from obtained nonlinear regression analysis. The concentration of inhibitor to cause 50% inhibition of original enzyme activity (IC_50_) was determined by nonlinear regression using Graphpad Prism software (London, UK). The percentage inhibition was calculated from *V*_m_ values.

## 4. Conclusions

In conclusion, an HPLC method was developed and validated for high throughput screening of compounds mediated by the CYP2C11 enzyme using salicylic acid as a tested inhibitor. Optimisation for in vitro CYP2C11 enzymatic reaction was achieved with regard to enzyme concentration and incubation time of the reaction. The amount of marker metabolite was quantified by a highly accurate and sensitive HPLC method. In this study, the in vitro CYP2C11 inhibition assay data demonstrated that Salicylic acid acts reversibly as a noncompetitive inhibitor, which may inhibit cytochrome P450 2C11 enzyme activity. The outcomes obtained provide useful tips for the effective use of salicylic acid in clinics. However, further in vitro and in vivo investigational studies are needed to determine the effect of salicylic acid administration in humans for full healthcare screening, regarding the safety of this drug when administrated concomitantly with regular medicines.

## Figures and Tables

**Figure 1 molecules-24-04294-f001:**
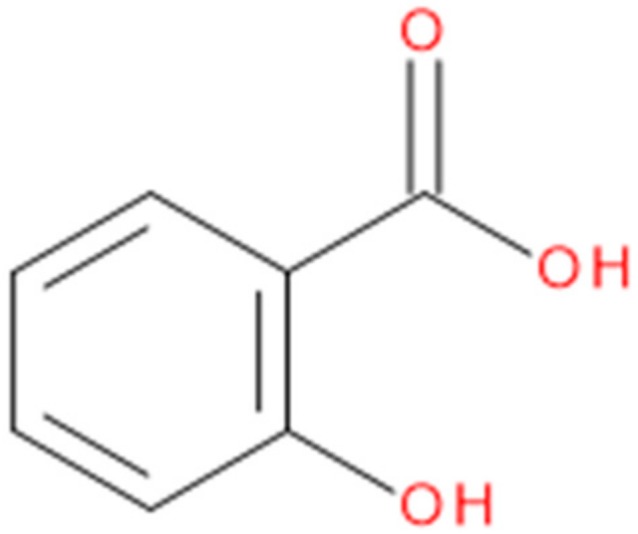
Structure of salicylic acid.

**Figure 2 molecules-24-04294-f002:**
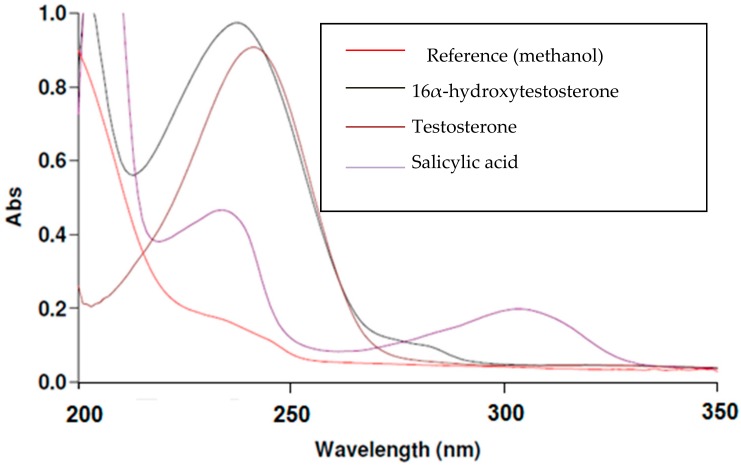
Overlain spectra of salicylic acid, testosterone and 16α-hydroxytestosterone components in the CYP2C11 assay.

**Figure 3 molecules-24-04294-f003:**
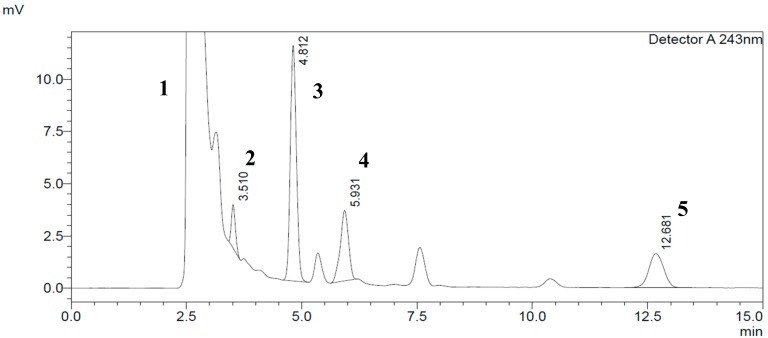
Typical HPLC chromatogram of CYP2C11 components in standard rat microsomal medium at 243 nm wavelength detection and a concentration of 150 µM testosterone. The peaks marked are: (**1**) NADPH (Nicotinamide Adenine Dinucleotide Phosphate Hydrogen)-regenerating system; (**2**) salicylic acid (100 µM); (**3**) phenacetin; (**4**) 16α-hydroxytestosterone; and (**5**) testosterone respectively.

**Figure 4 molecules-24-04294-f004:**
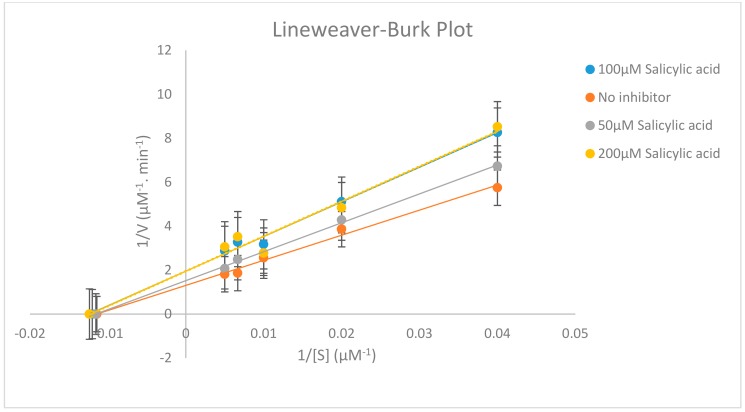
Lineweaver–Burk plot displaying the inhibition of CYP2C11 enzyme on the metabolism of testosterone into 16α-hydroxytestosterone using 0-200 µM salicylic acid. Each point represents averages of triplicate determinations.

**Figure 5 molecules-24-04294-f005:**
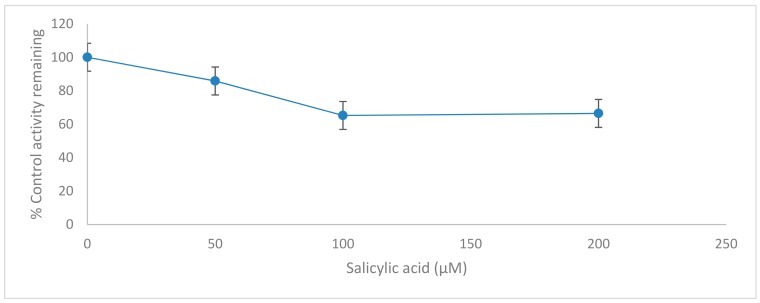
Inhibition of CYP2C11 isoform by salicylic acid (0–200 µM) in rat liver microsomes. Data represent averages of triplicates and are expressed as percentage of remaining control activity.

**Table 1 molecules-24-04294-t001:** Analytical performances.

Standards	Testosterone	16α-hydroxytestosterone
Regression equation	Y = 0.0189X − 0.0203	Y = 0.0204X − 0.0036
*r* ^2^	0.9999	0.9998
Linear range	10–400 µM	10–100 µM

**Table 2 molecules-24-04294-t002:** Limit of detection (LOD) and limit of quantitation (LOQ) for testosterone and 16α-hydroxytestosterone.

Standards	Testosterone	16α-hydroxytestosterone
Limit of Detection (LOD)	3.276 µM	1.350 µM
Limit of Quantitation (LOQ)	9.927 µM	4.914 µM

**Table 3 molecules-24-04294-t003:** Intra-assay variation for testosterone (n = 3).

Testosterone Standard	Mean Activity ^a^ (µM)	Standard Deviation	Relative Standard Deviation (%)
Low-activity standard (C = 25 µM)	22.8856	0.4851	2.1198
Medium-activity standard (C = 100 µM)	102.7028	0.6444	0.6275
High-activity standard (C = 200 µM)	201.2282	4.9342	2.4520

**^a^** Mean found concentration (µM).

**Table 4 molecules-24-04294-t004:** Intra-assay variation for 16α-hydrotestosterone (n = 3).

16α-hydroxytestosterone Standard	Mean Activity ^a^ (µM)	Standard Deviation	Relative Standard Deviation (%)
Low-activity standard (C = 10 µM)	8.5631	0.3848	4.4946
Medium-activity standard (C = 40 µM)	31.0608	0.3516	1.1322
High-activity standard (C = 80 µM)	67.9271	0.2154	0.3172

**^a^** Mean found concentration (µM).

**Table 5 molecules-24-04294-t005:** Inter-assay variation for testosterone.

Testosterone Standard (µM)		Mean Area Peak (n = 3 each Level)	Mean ^a^ Activity (µM)	Standard Deviation	Relative Standard Deviation (%)
Low-activity standard (C = 25 µM)	Day 1	0.7057	24.3194	0.6826	2.8068
	Day 2	0.5725			
	Day 3	0.8314			
Medium-activity standard (C = 100 µM)	Day 1	2.9393	101.7997	2.0495	2.0133
	Day 2	2.3360			
	Day 3	3.4765			
High-activity standard (C = 200 µM)	Day 1	5.9357	210.938	8.8989	4.2187
	Day 2	4.8596			
	Day 3	7.3105			

**^a^** Mean concentration (µM).

**Table 6 molecules-24-04294-t006:** Inter-assay performance of 16α-hydroxytestosterone.

16α-hydroxytestosterone Standard (µM)		Mean Area Peak (n = 3 each level)	Mean ^a^ Activity (µM)	Standard Deviation	Relative Standard Deviation (%)
Low-activity standard (C = 10 µM)	Day 1	0.1932	10.5557	0.7102	6.7286
	Day 2	0.3154			
	Day 3	0.1227			
Medium-activity standard (C = 40 µM)	Day 1	0.7802	38.1011	2.6567	6.9730
	Day 2	1.2850			
	Day 3	0.4912			
High-activity standard (C = 80 µM)	Day 1	1.6356	79.7765	2.4168	3.0295
	Day 2	2.6480			
	Day 3	1.1200			

**^a^** Mean concentration (µM).

**Table 7 molecules-24-04294-t007:** Stability test data for testosterone.

	Nominal Level (Actual Concentration of Testosterone (µM))			
		25	100	200
**Calculated concentration (µM)**	0 h	20.0423	80.1101	159
	24 h	21.6355	80.1101	159
	48 h	21.5520	86.0148	162.3135
	72 h	21.4054	80	163.3444
**% Recovery ^a^**	24 h	107.9492	100	100
	48 h	107.5323	107.3707	102.0839
	72 h	106.8007	99.8624	102.7323
**Accuracy ^b^ (%)**	0 h	119.8305	119.8898	120.5
	24 h	113.4576	119.8898	120.5
	48 h	113.7918	113.9851	118.8432
	72 h	114.3783	120	118.3277

Note: ^a^ % recovery = (concentration of testosterone at 24 h/standard concentration of testosterone) × 100; ^b^ Accuracy = 100 − ((calculated concentration − actual concentration)/actual concentration) × 100.

**Table 8 molecules-24-04294-t008:** Stability test data of 16-alfa hydroxytestosterone.

	Nominal Level (Actual Concentration of 16-alfa Hydroxytestosterone (µM))			
		10	40	80
**Calculated concentration (µM)**	0 h	10.533	38.0055	77.5659
	24 h	10.9725	41.3021	88.5549
	48 h	11.1923	44.2142	91.8516
	72 h	10.4	35.0866	76.5
**% Recovery ^a^**	24 h	109.725	103.2552	110.6936
	48 h	111.923	110.5355	114.8145
	72 h	104	87.7165	95.625
**Accuracy ^b^ (%)**	0 h	94.67	104.9862	103.0426
	24 h	90.275	96.7448	89.3064
	48 h	88.077	89.4645	85.1855
	72 h	96	112.2835	104.375

Note: ^a^ % recovery = (concentration of testosterone at 24 h/standard concentration of testosterone) × 100; ^b^ Accuracy = 100 − ((calculated concentration − actual concentration)/actual concentration) × 100.

**Table 9 molecules-24-04294-t009:** Retention time, peak area and resolution variation upon using two different mobile phase composition modes at a flow rate of 0.8 mL/min.

Mobile Phase Composition	Compounds	Average Retention Time (n = 3) (min)	Average Area Peak (n = 3) (Arb units)	Resolution
70% Methanol + 30% Phosphate buffer at pH = 3.36	Salicylic acid (100 µM)	3.468	63443	16α-hydroxytestosterone and phenacetin were well separated (good resolution) (Difference in retention time = 0.842 min).
	Phenacetin (50 µM)	4.512	143502	
	Testosterone (200 µM)	10.726	644120	
	16α- hydroxytestosterone (50 µM)	5.354	222620	
68% Methanol + 32% Phosphate buffer at pH = 3.36	Salicylic acid (100 µM)	3.572	67994	16α-hydroxytestosterone and phenacetin were well separated from each other (very good resolution) (Difference in retention time = 1.083 min).
	Phenacetin (50 µM)	4.730	141116	
	Testosterone (200 µM)	12.359	641325	
	16α- hydroxytestosterone (50 µM)	5.813	216239	

**Table 10 molecules-24-04294-t010:** Retention time and peak area variation using two different column temperatures.

Mobile Phase Composition	Compounds	Average Retention Time (n = 3) (min)	Average Area Peak (n = 3)	Resolution
68% Methanol + 32% Phosphate buffer at pH = 3.36 at a flow rate = 0.8 mL/min and T = 25 °C	Salicylic acid (100 µM)	3.549	50965	All compounds were well separated from each other. Difference in retention time between salicylic acid and phenacetin was 1.287 min.
	Phenacetin (50 µM)	4.836	135948	
	Testosterone (200 µM)	13.287	597882	
	16α- hydroxytestosterone (50 µM)	6.025	184493	
68% Methanol + 32% Phosphate buffer at pH = 3.36 at a flow rate= 0.8 mL/min and T = 30 °C	Salicylic acid(100 µM)	3.492	50536	All components were well separated from each other. Difference in retention time between salicylic acid and phenacetin was 1.26 min.
	Phenacetin (50 µM)	4.752	136505	
	Testosterone (200 µM)	12.472	598229	
	16α- hydroxytestosterone (50 µM)	5.854	175528	

**Table 11 molecules-24-04294-t011:** Pharmacokinetic parameters of CYP2C11 inhibition study. Values are expressed as mean ± SD (n = 3). Note: *p* < 0.0001.

Pharmacokinetic Parameters	No Inhibitor	50 µM Salicylic Acid	100 µM Salicylic Acid	200 µM Salicylic Acid
***K*_m_** **(µM)**	87.5613 ± 3.0516	86.5999 ± 3.0855	83.3333 ± 3.2064	80.8335 ± 3.3056
***V*_max_** **(µM^−1^∙min^−1^)**	0.7668 ± 0.1445	0.6581 ± 0.1684	0.5000 ± 0.2216	0.5094 ± 0.2175
***Cl*_int_** **(µM^−2^∙min^−1^)**	0.0087 ± 0.1228	0.0075 ± 0.1425	0.0060 ± 0.1782	0.0063 ± 0.1697
**α^’^**	-	1.1651 ± 0.1437	1.5336 ± 0.1092	1.5052 ± 0.1112
**% inhibition**	-	14.1766 ± 0.6661	34.7967 ± 0.2714	33.5642 ± 0.2814

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
