# Peer review of "Validation of an HPLC Method for the Simultaneous Quantification of Metabolic Reaction Products Catalysed by CYP2C11 Enzymes in Rat Liver Microsomes: In Vitro Inhibitory Effect of Salicylic Acid on CYP2C11 Enzyme"

_molecules, 2019, doi:10.3390/molecules24234294_

Round 1

Reviewer 1 Report

General comments

The manuscript presented by Salhab and co-authors describes an HPLC method to detect metabolites such as testosterone and 16α-hydroxytestosterone that are the substrate and product of the CYP2C11 enzyme, respectively. Besides, the authors also use the method developed to study the inhibitory effect of salicylic acid on the CYP2C11 enzyme. Although authors present in the manuscript a very comprehensive study to corroborate the method developed, the presentation, in the same manuscript, of the inhibitory effect of salicylic acid on the CYP2C11 enzyme and its potential use in clinic, makes it rather confusing. Authors should try to re-write or reformulate the manuscript in order to make it more clear regarding its objectives (an analytical method manuscript that uses the study of the inhibitory effect of salicylic acid on the CYP2C11 enzyme as an example, or the development of an analytical method to answer a clinical issue).

Questions to be addressed by authors:

Pg.3 – The baseline of the four spectra represented in Figure 2 should be corrected and the inset legend reformulated in order to allow its reading.

Pg3, L80 – Authors refer the 243nm as the “isoabsorptive point”. As the spectra showed in Figure 2 do not show similarities between the molar extinction coefficients of all the species, a better definition such as, “maximum of the absorption band”, should be considered.

Pg.3, L93 – To a better understanding of the manuscript, author should consider the revision of section 2.2.2.. The use of reference 13 is not clear as well as no comparison is performed with the study reported in it.

Pg.7, L176 – to a better understanding and robustness of the manuscript, authors should consider a revision of section 2.2.6.. The results are presented in two tables, table 9 and 10, and very few comments are done related with the results obtained. In fact, the comments to the results represented in table 10 refer to a “slight variation”. As this is also an analytic methods’ manuscript, the “slight variations” should be quantified and discussed.

Pg.9 – Figure 5 caption is not readable and should be revised.

Pg.9 – From the enzyme kinetics perspective, “no inhibitor” should be used instead of “negative control”.

Pg.10, L214 – Authors interpret the type of inhibition as “non-competitive” which is in accordance with the Lineweaver-Burk plot presented. However, from the analysis to Table 11, it seems that both KM and Vmax are reduced concomitantly with the higher amounts of salicylic acid, which would indicate an “uncompetitive” inhibition. This may be due to the error in the determination of the parameters, also possible to observe in Figure 5. Therefore, to avoid misinterpretation of the data, authors should present the errors of the kinetic parameters in Table 11, demonstrating that the KM values are comparable within the error.

In general terms, and to improve the quality of the document, authors should revise and edit all tables and figures, especially to the significant algarisms of the values presented.

Author Response

Firstly, thank you for taking time to review our research paper, and providing us with comments.

Point-by-point response:

General Comments:

The "General Objective". We agree; we have revised as you have recommended (An analytical method that uses the study of the inhibitory effect of salicylic acid on the CYP2C11 enzyme as a example- proof of concept study with a genuine clinical significance). See the attached manuscript below, (Lines 66-68).

Specific Comments:

Figure 2: Revised and legend was reconfigured, as recommended, see the attached manuscript below (Figure 2). The baseline for Figure 2 has also been corrected, as recommended.

Line 80: Revised and corrected. "isoabsorptive point" has been replaced by "maximum of the absorption band". See the manuscript attachment below (Lines 79-80).

Section 2.2.2. (Line 93): Revised with reference link now properly cited. LOD for both testosterone and 16-alpha hydroxytestosterone values were in the range of 1-4µM, which is consistent with the literature (Reference 13). See the attached manuscript below (Line 100).

Line 176: Revised and discussed. See the manuscript attachment below (Lines 192-196) & (Lines 204-210).

Figure 5 caption: Revised and corrected. See the manuscript attachment below (Lines 219-220).

Page 9: "No inhibitor" has now been replaced by "negative control" . see the manuscript attachment below (Figure 5 and Table 11).

Page 9: Revised and errors are now presented: See the manuscript attachment (Table 11) and (Lines 233).

We have undertaken editing with a view to improve the quality of the manuscript.

Reviewer 2 Report

The authors aimed to develop and validate a HPLC method to quantitate the hydroxylized testosterone formed upon CYP2C11 catalyzed oxidation of testosterone in vitro. In addition, the method was used to study the inhibitory effect of salicylic acid on the CYP2C11 enzyme.

The manuscript cannot be recommended for publication in Molecules due to severe drawbacks.

The work is lacking novelty. A HPLC method to quantitate hydroxytestosterones formed in liver microsomes in vitro was developed and validated by Li and Letcher back in 2002 (J. Chromatogr Sci, 40, 2002). Also, the inhibitory potential of salicylic acid on the CYP2C11 enzyme is known. The results of the validation are not convincing. How can a linearity be determined in the range of 0-100 µM when the LOD is above 1 µM? Can the authors really determine exact amounts in the range of nM with the method (e.g. LOD 3.276 µM). The peak assigned to 16alpha-hydroxytestosterone seems to hide another peak. How can the authors be sure that no other matrix constituents coelute with the hydroxylated testosterone? The intra assay shows huge deviations from the accurate concentration (e.g. 31 versus 40 µM). Also, the inter-assay experiments shows huge variations (e.g. mean peak area of 0.12 to 0.32). The stability tests reveals low accuracy and high deviation (20%) of the calculated concentration from the used concentration. For preparing the standards, the authors weighed 5.7684 mg testosterone. An accuracy of 0.1 µg is pretty hard to obtain with a standard balance. The results were not discussed at all. A native speaker needs to revise the manuscript. Figure 2: legend is partially not visible line 198: 200 µM of what?

Author Response

Firstly, we would like to thank the reviewer for taking time to review our research paper, and providing us with comments.

Point-by-point response:

Lack of novelty. As we stated in our introduction section, various papers have reported the pharmacokinetics and metabolism of Salicylic acid (Reference 9), but up till the present, there is no study that shows the in vitro inhibitory effect of Salicylic acid on the CYP2C11 enzyme activity (to our knowledge). Our HPLC method for CYP2C11 assay was validated according to ICH guidelines. All parameters (% error, standard deviation, Precision, Accuracy, LOD, and LOQ) were met, as stated in the discussion section of the manuscript.

The paper you have suggested (Li and Letcher 2002) highlights the HPLC development of testosterone and its all metabolites only (6-alfa, 15-alfa, 6-beta, 7-alfa, 16-alfa, 16-beta, 2-alfa and 11-beta hydroxytestosterone) and quantifying the amount of each product formed by conducting in vitro metabolism of testosterone in Gray Seal microsomes, (a negative control experiment was conducted without using a tested inhibitor). In our paper, an HPLC method was developed and validated for CYP2C11 assay using salicylic acid as a tested inhibitor for the urgent systematic investigation of Salicylic acid on Cytochrome P450. Ours is thus a completely different objective to this paper. We are sorry if this wasn't clear and we have subsequenstly altered our objective statement (Lines 66-68).

Thank-you for pointing the linear range typo errors out. The calibration curve of 16-alfa hydroxytestosterone was run at 10, 20, 40, 60, 80 and 100µM, and a blank solution was run on HPLC instrument, which only consisted of internal standard (Phenacetin). So the linear range of Testosterone is (10-400µM) (revised and corrected) and the linear range of 16-alfa hydroxytestosterone (10-100µM) (revised and corrected). Therefore, LOD values for both compounds are now lower than the linear range of the two compounds. See the manuscript attachment below (Table 1).

LOD and LOQ exact amounts were calculated from both equations in section 2.2.2. (see the manuscript attachment below).

Each of the components were run separately on HPLC and the retention time of 16-alpha hydroxytestosterone was at 5.95 mins. We have seen no evidence to suggest there is any other compound co-eluting with this peak and the fact that ICH guidelines were met in vaildation indirectly supports our theory. Ultimately, a LC-MS scan of the elution peak would confirm this.

Intra-assay variation. This has now been revised and clarified. A calibration curve was run for intra-assay variation at 10,20, 60, 80 and 100µM concentrations (y=0.0204x-0.0036, R2=0.9998). The calculated concentration for three actual concentrations were carried out in triplicate from the calibration curve. The % error was less than 5% which meets the ICH guidelines.

Inter-assay variation (0.12 to 0.32). This has now been revised and clarified. Mean area peak for Day 2 was calculated from the calibration curve of Day 2, and mean area peak for Day 3 was calculated from the calibration curve of day 3. A new calibration curve was run each day (1,2 and 3), so different calibration curve equations were obtained for days 1,2, and 3. Therefore the % error was calculated as 6.73%, which is less than 10% and thus meets guidelines.

Stability test data of 16-alfa hydroxytosterone. This has been revised and clarified. The calculated concentrations were calculated from the average calibration curve equation, %recovery and accuracy and met the ICH guidelines and low deviation from the used concentration. See the manuscript attachment (Table 8).

Stability test data of testosterone. This has been revised and clarified. The calculated concentrations were calculated from the average calibration curve equation of Days 1-4, % error, and recovery were within the ICH guidelines (see Table 7 in the manuscript attachment).

Preparing the standard. This has been revised and corrected. See the manuscript attachment (Line 311-312).

Results have been fully discussed (see the manuscript attachment below discussion section).

The manuscript has been reviewed by native speakers.

Figure 2: This has been revised and corrected. See the maniscript attachment below (Figure 2).

Line 198: This has been revised and corrected. See the manuscript attachment below (Line 213).

We think that by making these corrections and revisions that you have suggested the paper is now of a much better standard and we hope that our study objectives and analytical validation are now much clearer.

Reviewer 3 Report

This paper discusses validation of HPLC method for determination testosterone, its metabolite, phenancetin and salicylic acid. The method was validated according to standard validation parameters and it satisfied ICH criteria therefore it can be used for analysis of CYP2C11 in vitro assay. Other part of the paper deals with inhibitory effect of salicylic acid on CYP2C11 enzyme.

In the title it is mentioned that the method was developed but authors state that they used only one type of column and one mobile phase system in 2 concentrations. Either the data of the method development should be presented and discussed or the word “development” should be omitted from the title.

Comments:

Line 24: drug – drug interactions Figure 2 – in this preview the legend is not clearly seen Figure 3 – it would be advisable to put number in the brackets after each compound, eg salicylic acid (2). It is easier to follow the figure. Line 123: illustrate Line 133: illustrate Table 5 and 6: parameter Mean activity has superscript which is not explained after each table Lines 141 and 161: were samples kept in the dark? Table 7: unify number of digits Lines 150 – 151 and 168 – 169: why were calibration curves for testosterone presented as average and for the metabolite not? Table 8: unify number of digits Line 187: add: considering change in mobile phase ratio Line 195: add: considering change in column temperature Line 198: there something missing from the sentence, presence of what? Lines 216 and 218: the discussion should be more advanced. Explain what does it mean when the Km value remains unaffected and what does the chang in Vm mean? Define Vm Line 234: Cl Line 236: the details concerning HPLC column used should be mentioned in the Experimental part Line 256: good resolution of investigated compounds Line 273: a volume of 10 µL of dissolved supernantant Line 316: high throughput

Author Response

Firstly, we would like to thank you for taking time to review our research paper and provide us with comments.

Point-by-point response:

"Development" word is omitted from the title as you recommended, since only one mobile phase system was used (68% Methanol+32% Phosphate buffer at pH=3.36) and only one type of column been used (SUPELCO 25cm×4.6mm×5μm).

Line 24: revised and corrected (drug-drug interactions) (see the manuscript attachment below) (Line 25).

Figure 2: the legend has been corrected (see the manuscript attachment below).

Figure 3: revised and corrected (a number been placed before each compound) (see the manuscript attachment below).

Line 123 and 133: revised and corrected "illustrate" (see the manuscript attachment below) (Lines 123,133).

Table 5 and 6: revised and corrected. The parameter mean activity superscript has been explained after Table 5 and 6 (see the manuscript attachment below).

Lines 141 and 161: revised and clarified. The stability test of testosterone and 16-alfa hydroxytestosterone investigated for three different concentrations stored for 72 hours and kept at room temperature in natural light condition (see the manuscript attachment below) (Lines 144 and 165).

Table 7: revised and  corrected. The numbers has been rounded to four decimal places (see the manuscript attachment below).

Lines 150-151: revised and corrected. The calibrartion curves for testosterone were averaged and the calculated concentrations were calculated from the average calibration curve of testosterone. See the stability test section for 16-alfa hydroxytestosteone in the manuscript attachment below (Line 153).

Lines 168-169: revised and corrected. The calibration curves for 16-alfa hydroxytestosterone were averaged and the calculated concentrations of the metabolite at different concentrations were calculated from the average calibration curve of the metabolite of day 1-4 (see the manuscript attachment below) (Line 173-175).

Table 8: revised and corrected. see the manuscript attachment below.

Line 187: revised and added. See the manuscript attachment below (Line 198).

Line 195: revised and corrected. See the manuscript below (Line 198).

Line 198: revised and added. "Different concentrations of testosterone (200, 150,100, 50 and 25µM) were incubated in the presence of 0,50, 100 and 200µM) of salicylic acid. See the manuscript attachment below (Line 213).

Lines 216 and 218: revised and discussed:Vmax and Km have been defined. See the manuscript attachment below (Lines 231-238).

Line 234: revised and corrected. See the manuscript attachment below (Line 254).

Line 236: revised and added. See the manuscript attachment below (263-264).

Line  256: revised and corrected. See the manuscript attachment below (Line 277).

Line 273: revised and corrected. See the manuscript attachment below (Line 294).

Line 316: revised and corrected. See the manuscript attachment below (Line 337).

Reviewer 4 Report

This paper describes the development and validation of HPLC method for high-throughput screening compounds mediated by CYP2C11 enzyme using salicylic acid as an inhibitor. The in vitro CYP2C11 inhibition assay data demonstrated that Salicylic acid acts potently as reversibly non-competitive inhibitor. This study could potentially be useful for application of  salicylic acid in clinic. Although further studies are needed, I believe this study will be appealing to the broad readership of Molecules. I suggest publishing in Molecules after some minor revisions:

For changing the column temperature, the experiments were run at 25 degree C and 30 degree C. It wasn’t mentioned why these two temperatures were selected in the paper. More details are needed as to why these two Temperatures. Why not conduct more experiments on different  temperatures that  lower than 25 degree C or and higher than 30 degree C. For the stability test of testosterone, it was mentioned that three different concentrations – 25, 100, and 200 μ Is that a standard as to why these concentrations? While the stability test of 16α-hydroxytestosterone use three different concentrations  10, 40, and 80 μM.

Author Response

Firstly, we would like to thank you for taking time to review our research paper and providing us with comments.

Point-by-point response:

Changing column temperature: 25ºC and 30ºC. This has been revised and clarified. The method was optimized using 30ºC as the HPLC column temperature, thus 25ºC was selected as it is within 5ºC of this and this is specified in robustness test guidelines. In addition, experiments were run at 25°C and 30ºC in order for the results to be more applicable for the in vivo studies of the inhibitory effect of Salicylic acid for routine analysis at ambient temperatures in clinic. See the manuscript attachment below (Lines 199-207).

Why not conduct more experiments at different temperatures?

This has been revised and clarified. The robustness test is a part of method validation, which is performed during method optimization, Because the method had been optimized at 30°C, so 25°C experiment was conducted to see the influence of retention time and area peak of each compound up to 5°C of the optimized temperature, as the robustness test requires. More temperatures could have been chosen but this would exceed the necesary requirements of the validation.

Why stability test of Testosterone conducted at three different concentrations (25, 100, and 200µM)?

This has been revised and clarified. Three different concentrations of testosterone were chosen as low, medium and high concentrations (25, 100 and 200µM), approximating to its Km value in human liver microsomes (E. V.A., et al., 1998). A range of substrate concentrations (25-200µM) were chosen to conduct the inhibitory effect of Salicylic acid on CYP2C11 enzyme activity as this range of concentration represents  an  approximation to the Km value of testosterone in human liver (E. V.A., et al., 1998). See the attached manuscript below (Line 143 and Reference 14).

Why does the stability test of 16-alfa hydroxytestosterone use three different concentrations (10, 40, and 80µM)?

This has been revised and clarified: A calibration curve of 16-alfa hydroxytestosterone was run upon the quantification of the product formed in the range of 10-100µM, mainly because the negative control (no inhibitor) experiment that we conducted showed that testosterone at 200µM metabolised to 16-alfa hydroxytestosterone at its maximal level (35µM), so its important to investigate the stability of 16-alfa hydroxytestosterone on 3 different ranges of 16-alfa hydroxytestosterone (Lower, equal, and higher ) concentration than 35µM (supplementary materials). The purpose of conducting stability tests at three different concentrations is because the metabolite calibration standards are prepared beforehand. It is also quite common to do this in validation to ICH guidelines.

Round 2

Reviewer 2 Report

The authors substantially improved their manuscript by revising the validation results and more clearly stating the objective of their study. All comments raised by the reviewer were considered by the authors. Thus, publication of the manuscript in its present form can be recommended.

Reviewer 3 Report

The values in Tables 7 and 8 have not been corrected. For example: if one number is presented as 86.0148 then 80 should be presented as 80.0000, that means to unify the number of digits in the table. Write all the numbers with the same number of digits.